# ROBUST VISUAL DOMAIN RANDOMIZATION FOR REINFORCEMENT LEARNING

## ABSTRACT

Producing agents that can generalize to a wide range of visually different environments is a significant challenge in reinforcement learning. One method for overcoming this issue is visual domain randomization, whereby at the start of each training episode some visual aspects of the environment are randomized so that the agent is exposed to many possible variations. However, domain randomization is highly inefficient and may lead to policies with high variance across domains. Instead, we formalize the visual domain randomization problem, and show that minimizing the policy's Lipschitz constant with respect to the randomization parameters leads to low variance in the learned policies. We propose a regularization method where the agent is only trained on one variation of the environment, and its learned state representations are regularized during training to minimize this constant. We conduct experiments that demonstrate that our technique leads to more efficient and robust learning than standard domain randomization, while achieving equal generalization scores.

## 1    INTRODUCTION

Deep Reinforcement Learning (RL) has proven very successful on complex high-dimensional problems ranging from games like Go (Silver et al., 2017) and Atari games (Mnih et al., 2015) to robot control tasks (Levine et al., 2016). However, one prominent issue is that of overfitting, illustrated in figure 1: agents trained on one domain fail to generalize to other domains that differ only in small ways from the original domain (Sutton, 1996; Cobbe et al., 2018; Zhang et al., 2018b; Packer et al., 2018; Zhang et al., 2018a; Witty et al., 2018; Farebrother et al., 2018). Good generalization is essential for problems such as robotics and autonomous vehicles, where the agent is often trained in a simulator and is then deployed in the real world where novel conditions will certainly be encountered. Transfer from such simulated training environments to the real world is known as crossing the *reality gap* in robotics, and is well known to be difficult, thus providing an important motivation for studying generalization.

We focus on the problem of generalizing between environments that visually differ from each other, for example in color or texture, but where the underlying dynamics are the same. In reinforcement learning, prior work to address this topic has studied both *domain adaptation* and *domain randomization*. Domain adaptation techniques aim to update the data distribution in simulation to match the real distribution through some form of canonical mapping or using regularization methods (James et al., 2018; Bousmalis et al., 2017; Gamrian & Goldberg, 2018). Alternatively, domain randomization, in which the visual and physical properties of the training domains are randomized at the start of each episode during training, has also been shown to lead to improved generalization and transfer to the real world with little or no real world data (Tobin et al., 2017; Sadeghi & Levine, 2016; Antonova et al., 2017; Peng et al., 2017; Mordatch et al., 2015; Rajeswaran et al., 2016; OpenAI, 2018). However, domain randomization has been empirically shown to often lead to suboptimal policies with high variance in performance over different randomizations (Mehta et al., 2019). This issue can cause the learned policy to underperform in any given target domain.

We propose a regularization method for learning policies that are robust to irrelevant visual changes in the environment. Our work combines aspects from both domain adaptation and domain randomization, in that we maintain the notion of randomized environments but use a regularization method to achieve good generalization over the randomization space. Our contributions are the following:

- We formalize the visual domain randomization problem, and show that the Lipschitz constant of the agent's policy over visual variations provides an upper bound on the agent's robustness to these variations.

- We propose an algorithm whereby the agent is only trained on one variation of the environment but its learned representations are regularized so that the Lipschitz constant is minimized.

- We experimentally show that our method is more efficient and leads to lower-variance policies than standard domain randomization, while achieving equal or better returns and generalization ability.

This paper is structured as follows. We first review related work, formalize the visual generalization problem, and present our theory contributions. We then describe our regularization method, and illustrate its application to a toy gridworld problem. Finally, we compare our method with standard domain randomization and other regularization techniques in complex visual environments.

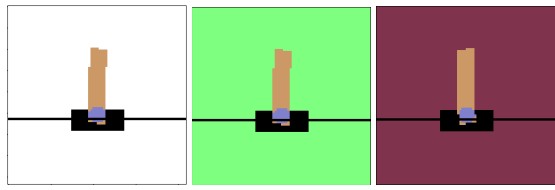

Figure 1: Illustration of the visual generalization challenge in reinforcement learning. In this cartpole domain, the agent must learn to keep the pole upright. However, changes in the background color can completely throw off a trained agent.

## 2    RELATED WORK

### 2.1    GENERALIZATION IN DEEP REINFORCEMENT LEARNING

Generalization to novel samples is well studied in supervised learning, where evaluating generalization through train/test splits is ubiquitous. However, evaluating for generalization to novel conditions through such train/test splits is not common practice in Deep RL. Zhang et al. (2018b) show that Deep RL algorithms are shown to suffer from overfitting to training configurations and to memorize training scenarios in discrete maze tasks. Packer et al. (2018) study performance under train-test domain shift by modifying environmental parameters such as robot mass and length to generate new domains. Farebrother et al. (2018) propose using different game modes of Atari games to measure generalization. They turn to supervised learning for inspiration, finding that both L2 regularization and dropout can help agents learn more generalizable features. These works all show that standard Deep RL algorithms tend to overfit to the environment used during training, hence the urgent need for designing agents that can generalize better.

### 2.2    DOMAIN RANDOMIZATION

We distinguish between two types of domain randomization: visual randomization, in which the variability between domains should not affect the agent's policy, and dynamics randomization, in which the agent should learn to adjust its behavior to achieve its goal. Visual domain randomization, which we focus on in this work, has been successfully used to directly transfer RL agents from simulation to the real world without requiring any real images (Tobin et al., 2017; Sadeghi & Levine, 2016; Kang et al., 2019). These approaches used low fidelity rendering and randomized scene properties such as lighting, textures, camera position, and colors, which led to improved generalization.

Other work has also combined domain randomization and domain adaptation techniques (James et al., 2018; Chebotar et al., 2018; Gamrian & Goldberg, 2018). These approaches both randomize the simulated environment and penalize the gap between the trajectories in the simulations and the real world, either by adding a term to the loss, or learning a mapping between the states of the simulation and the real world. However, these methods require a large number of samples of real world trajectories, which can be expensive to collect.

Prior work has, however, noted the inefficiency of domain randomization. Mehta et al. (2019) show that domain randomization may lead to suboptimal policies that vary a lot between domains, and propose to train on the most informative environment variations within the given randomization ranges. Zakharov et al. (2019) also guide the domain randomization procedure by training a *DeceptionNet*, that learns which randomizations are actually useful to bridge the domain gap for image classification tasks.

## 2.3 LEARNING DOMAIN-INVARIANT FEATURES AND DOMAIN ADAPTATION

Learning domain-invariant features has emerged as a promising approach for taking advantage of the commonalities between domains. For instance, in the semi-supervised context, Bachman et al. (2014); Sajjadi et al. (2016); Coors et al. (2018); Miyato et al. (2018); Xie et al. (2019) enforce that predictions of their networks be similar for original and augmented data points, with the objective of reducing the required amount of labelled data for training. Our work extends such methods to reinforcement learning.

In the reinforcement learning context, several other papers have also explored this topic. Tzeng et al. (2015) and Gupta et al. (2017) add constraints to encourage networks to learn similar embeddings for samples from both a simulated and a target domain. Daftry et al. (2016) apply a similar approach to transfer policies for controlling aerial vehicles to different environments. Bousmalis et al. (2017) compare different domain adaptation methods in a robot grasping task, and show that they improve generalization. Wulfmeier et al. (2017) use an adversarial loss to train RL agents in such a way that similar policies are learned in both a simulated domain and the target domain. While promising, these methods are designed for cases when simulated and target domains are both known, and cannot straightforwardly be applied when the target domain is only known to be within a distribution of domains.

Concurrently and independently of our work, Aractingi et al. (2019) also propose a regularization scheme to learn policies that are invariant to randomized visual changes in the environment without any real world data. Our work differs from theirs in that we propose a theoretical justification for this regularization and an analysis of the effects of this regularization on the learned representations. Crucially, whereas Aractingi et al. (2019) propose regularizing the network outputs, we regularize intermediate layers instead. In the appendix, we experimentally compare their regularization to ours and show that regularizing the network outputs leads to an undesirable trade-off between agent performance and generalization.

## 3 PROBLEM FORMULATION

We consider Markov decision processes (MDP) defined by $(\mathcal{S}, \mathcal{A}, R, T, \gamma)$, where $\mathcal{S}$ is the state space, $\mathcal{A}$ the action space, $R : \mathcal{S} \times \mathcal{A} \to \mathbb{R}$ the reward function, $T : \mathcal{S} \times \mathcal{A} \to Pr(\mathcal{S})$ the transition dynamics, and $\gamma$ the discount factor. In reinforcement learning, an agent's objective is to find a policy $\pi$ that maps states to distributions over actions such that the cumulative discounted reward yielded by its interactions with the environment is maximized.

### 3.1 VISUAL DOMAIN RANDOMIZATION

We consider a framework in which we are given a set of $N$ parameters that can be changed to visually modify the environment, defined within a *randomization space* $\Xi \subset \mathbb{R}^N$. These parameters can for example control textures, colors, or lighting. Denoting $J(\pi, \xi)$ the cumulative returns of a policy $\pi$, the goal is to solve the optimization problem defined by $J(\pi^*) = \max_\pi \mathbb{E}_\xi[J(\pi, \xi)]$.

Standard domain randomization, in which parameters $\xi$ are randomly sampled at the start of each training episode, empirically produces policies with strongly varying performance over different regions of the randomization space, as demonstrated by Mehta et al. (2019). This high variance can cause the learned policy to underperform in any given target domain. To yield insight into the robustness of policies learned by domain randomization, we start by formalizing the notion of a visually randomized MDP.

**Definition 1** *Let $M = (\mathcal{S}, \mathcal{A}, R, T, \gamma)$ be an MDP. A randomizer function of $M$ is a mapping $\phi : \mathcal{S} \to \mathcal{S}'$ where $\mathcal{S}'$ is a new set of states. The randomized MDP $M_\phi = (\mathcal{S}_\phi, \mathcal{A}_\phi, R_\phi, T_\phi, \gamma_\phi)$ is*

*defined as, for $s, s' \in \mathcal{S}$, $a \in \mathcal{A}$ :*

$$\mathcal{S}_\phi = \phi(\mathcal{S}), \quad \mathcal{A}_\phi = \mathcal{A}, \quad T_\phi(\phi(s')|\phi(s), a) = T(s'|s, a), \quad R_\phi(\phi(s), a) = R(s, a), \quad \gamma_\phi = \gamma$$

*Given a policy $\pi$ on MDP $M$ and a randomization $M_\phi$, we also define the agent's policy on $M_\phi$ as $\pi^\phi(\cdot|s) = \pi(\cdot|\phi(s))$.*

Despite all randomized MDPs sharing the same underlying rewards and transitions, the agent's policy can vary between domains. For example, in policy-based algorithms (Williams, 1992), if there are several optimal policies then the agent may adopt different policies for different $\phi$. Furthermore, for value-based algorithms such as DQN (Mnih et al., 2015), two scenarios can lead to there being different policies for different $\phi$. First, the (unique) optimal Q-function may correspond to several possible policies. Second, imperfect function approximation can lead to different value estimates for different randomizations and thus to different policies. To compare the ways in which policies can differ between randomized domains, we introduce the notion of Lipschitz continuity of a policy over a set of randomizations.

**Definition 2** *We assume the state space is equipped with a distance metric. A policy $\pi$ is Lipschitz continuous over a set of randomizations $\{\phi\}$ if for all randomizations $\phi_1$ and $\phi_2$ in $\{\phi\}$,*

$$K_\pi = \sup_{\phi_1, \phi_2 \in \{\phi\}} \sup_{s \in \mathcal{S}} \frac{D_{TV}(\pi(\cdot|\phi_1(s))\|\pi(\cdot|\phi_2(s)))}{|\phi_1(s) - \phi_2(s)|}$$

*is finite. Here, $D_{TV}(P\|Q)$ is the total variation distance between distributions (given by $\frac{1}{2}\sum_{a \in \mathcal{A}}|P(a) - Q(a)|$ when the action space is discrete).*

The following inequality shows that this Lipschitz constant is crucial in quantifying the robustness of RL agents over a randomization space. The smaller the Lipschitz constant, the less a policy is affected by different randomization parameters. Informally, if a policy is Lipschitz continuous over randomized MDPs, then small changes in the background color in an environment will have a small impact on the policy.

**Proposition 1** *We consider an MDP $M$ and a set of randomizations $\{\phi\}$ of this MDP. Let $\pi$ be a $K$-Lipschitz policy over $\{\phi\}$. Suppose the rewards are bounded by $r_{\max}$ such that $\forall a \in \mathcal{A}, s \in \mathcal{S}, |r(s, a)| \le r_{\max}$. Then for all $\phi_1$ and $\phi_2$ in $\{\phi\}$, the following inequalities hold :*

$$|\eta_1 - \eta_2| \le 2r_{\max} \sum_{t}^{\infty} \gamma^t \min(1, (t+1)K_\pi\|\phi_1 - \phi_2\|_\infty) \le \frac{2r_{\max}K_\pi}{(1-\gamma)^2}\|\phi_1 - \phi_2\|_\infty \quad (1)$$

*Where $\eta_i$ is the expected cumulative return of policy $\pi^{\phi_i}$ on MDP $M_{\phi_i}$, for $i \in \{1, 2\}$, and $\|\phi_1 - \phi_2\|_\infty = \sup_{s \in \mathcal{S}}|\phi_1(s) - \phi_2(s)|$.*

*Proof.* See appendix.

These inequalities shows that the smaller the Lipschitz constant, the smaller the maximum variations of the policy over the randomization space can be. In the following, we present a regularization technique that produces low-variance policies over the randomization space by minimizing the Lipschitz constant of the policy.

## 4 PROPOSED REGULARIZATION

We propose a simple regularization method to produce an agent with policies that vary little over randomized environments, despite being trained on only one environment. We start by choosing one variation of the environment on which to train an agent with a policy $\pi$ parameterized by $\theta$, and during training we minimize the loss

$$\mathcal{L}(\theta) = \mathcal{L}_{RL}(\theta) + \lambda \mathop{\mathbb{E}}_{s \sim \pi} \mathop{\mathbb{E}}_{\phi} \|f_\theta(s) - f_\theta(\phi(s))\|_2^2 \quad (2)$$

where $\lambda$ is a regularization parameter, $\mathcal{L}_{RL}$ is the loss corresponding to the chosen reinforcement learning algorithm, the first expectation is taken over the distribution of states visited by the current

policy which we assume to be fixed when optimizing this loss, and $f_\theta$ is a feature extractor used by the agent's policy. In our experiments, we choose the output of the last hidden layer of the value or policy network as our feature extractor. Minimizing the second term in this loss function minimizes the Lipschitz constant as defined above over the states visited by the agent, and causes the agent to learn representations of states that ignore variations caused by the randomization.

Our method can be applied to many RL algorithms, since it involves simply adding an additional term to the learning loss. In the following, we experimentally demonstrate applications to both value-based and policy-based reinforcement learning algorithms. Implementation details can be found in the appendix, and the code will be made available online.

## 5 EXPERIMENTS

### 5.1 ILLUSTRATION ON A GRIDWORLD

We first conduct experiments on a simple gridworld to illustrate the theory described above.

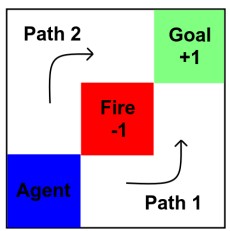
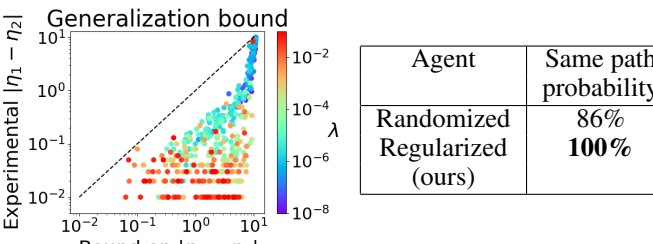

| Agent | Same path probability |
|---|---|
| Randomized | 86% |
| Regularized (ours) | **100%** |

Figure 2: Left: a simple gridworld, in which the agent must make its way to the goal while avoiding the fire. Center: empirical differences between regularized agents' policies on two randomizations of the gridworld compared to our theoretical bound in equation 1 (the dashed line). Each point corresponds to one agent, and 20 training seeds per value of $\lambda$ are shown here. Right: probability that different agents choose the same path for two randomizations of this domain. Our regularization method leads to more consistent behavior.

The environment we use is the $3 \times 3$ gridworld shown in figure 2, in which two optimal policies exist. The agent starts in the bottom left of the grid and must reach the goal while avoiding the fire. The agent can move either up or right, and in addition to the rewards shown in figure 2 receives -1 reward for invalid actions that would case it to leave the grid. We set a time limit of 10 steps and $\gamma = 1$. We introduce randomization into this environment by describing the state observed by the agent as a tuple $(x, y, \xi)$, where $(x, y)$ is the agent's position and $\xi$ is a randomization parameter with no impact on the underlying MDP. For this toy problem, we consider only two possible values for $\xi$: $+5$ and $-5$. The agents we consider use the REINFORCE algorithm (Sutton et al., 2000) with a baseline (see appendix), and a multi-layer perceptron as the policy network.

First, we observe that even in a simple environment such as this one, a randomized agent regularly learns different paths for different randomizations (figure 2). An agent trained only on $\xi = 5$ and regularized with our technique, however, consistently learns the same path regardless of $\xi$. Although both agents easily solve the problem, the variance of the randomized agent's policy can be problematic in more complex environments in which identifying similarities between domains and ignoring irrelevant differences is important.

Next, we compare the measured difference between the policies learned by regularized agents on the two domains to the smallest of our theoretical bounds in equation 1, which in this simple environment can be directly calculated. For a given value of $\lambda$, we train a regularized agent on the reference domain. We then measure the difference in returns obtained by this agent on the reference and on the regularized domain, and this return determines the agent's position along the $x$ axis. We then numerically calculate the Lipschitz constant from the agent's action distribution over all states, and use this constant to calculate the bound in proposition 1. This bound determines the agent's position along the $y$ axis. Our results for different random seeds and values of $\lambda$ are shown in figure 2. We

observe that increasing $\lambda$ does lead to decreases in both the empirical difference in returns and in the theoretical bound.

## 5.2 VISUAL CARTPOLE WITH DQN

We compare standard visual domain randomization to our regularization method on a more challenging visual environment, in terms of 1) training stability, 2) returns and variance of the learned policies, and 3) state representations learned by the agents.

### 5.2.1 EXPERIMENTAL SETTING

To run domain randomization experiments, we use a visual Cartpole environment shown in figure 1, where the states consist of raw pixels of the images. The agent must keep a pole upright as long as possible on a cart that can move left or right. The episode terminates either after 200 time steps, if the cart leaves the track, or if the pole falls over. The randomization consists of changing the color of the background. Each randomized domain $\xi \in \Xi$ corresponds to a color $(r, g, b)$, where $0 \leq r, g, b \leq 1$. Our implementation of this environment is based on the OpenAI Gym (Brockman et al., 2016). For training, we use the DQN algorithm with a CNN architecture similar to that used by Mnih et al. (2015). In principle, such a value-based algorithm should learn a unique value function independently of the randomization parameters we consider. However, as we will show function approximation errors cause different value functions to be learned for different background colors.

We compare the performance of three agents. The **Normal** agent is trained on only one domain (with a white background). The **Randomized** agent is trained on a chosen randomization space $\Xi$. The **Regularized** agent is trained on a white background using our regularization method with respect to randomization space $\Xi$. The training of all three agents is done using the same hyperparameters, and over the same number of steps.

### 5.2.2 PERFORMANCE DURING TRAINING

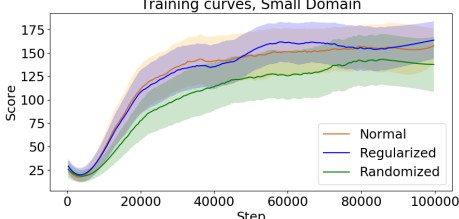 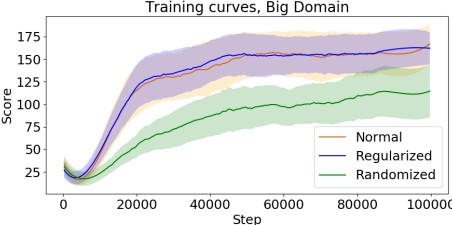

Figure 3: Training curves over randomization spaces $\Xi_{small}$ (left) and $\Xi_{big}$ (right). Shaded areas indicate the 95% confidence interval of the mean, obtained over 10 training seeds.

We first compare the performance of our agents during training. We train all three agents over two randomization spaces (environments with different background colors), having the following sizes :

- $\Xi_{small} = \{(r, g, b), 0.5 \leq r, g, b \leq 1.\} = [0.5, 1]^3 : \frac{1}{8}$ of the unit cube.

- $\Xi_{big} = [0, 1] \times [0.5, 1] \times [0, 1]$ : half the unit cube.

We obtain the training curves shown in figure 3. We find that the normal and regularized agents have similar training curves and are not affected by the size of the randomization space. However, the randomized agent learns more slowly on the small randomization space $\Xi_{small}$ (left), and also achieves worse performance on the bigger randomization space $\Xi_{big}$ (right). In high-dimensional problems, we would like to pick the randomization space $\Xi$ to be as large as possible to increase the chances of transferring to the target domain. We find that standard domain randomization scales poorly with the size of the randomization space $\Xi$, whereas our regularization method is more robust to a larger randomization space.

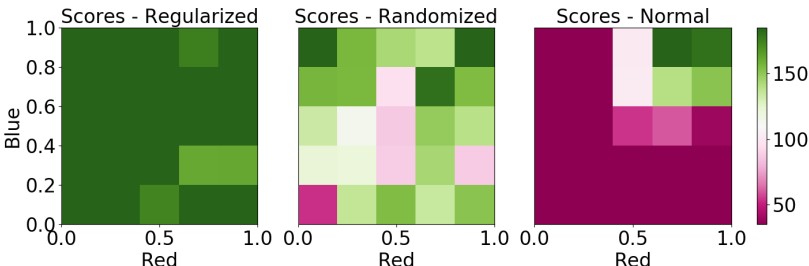

Figure 4: Comparison of the average scores of different agents over different domains. The scores are calculated over a plane of the (r,g,b) cube in $\Xi_{big}$, where $g = 1$ is fixed, averaged over 1000 steps. The training domain for both the regularized and normal agents is located at the top right. The regularized agent learns more stable policies than the randomized agent over these domains.

### 5.2.3 GENERALIZATION AND VARIANCE

We compare the returns of the policies learned by the agents in different domains within the randomization space. We select a plane within $\Xi_{big}$ obtained by varying only the R and B channels but keeping G fixed. We plot the scores obtained on this plane in figure 4. We see that despite having only been trained on one domain, the regularized agent achieves consistently high scores on the other domains. On the other hand, the randomized agent's policy exhibits returns with high variance between domains, which indicates that different policies were learned for different domains.

### 5.2.4 REPRESENTATIONS LEARNED BY THE AGENTS

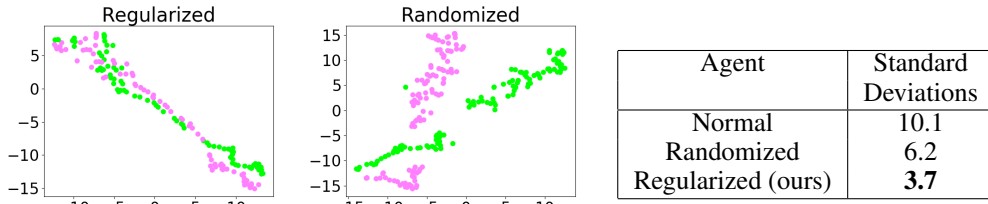

| Agent | Standard Deviations |
|---|---|
| Normal | 10.1 |
| Randomized | 6.2 |
| Regularized (ours) | **3.7** |

Figure 5: Left: Visualization of the representations learned by the agents for pink and green background colors and for the same set of states. We observe that the randomized agent learns different representations for the two domains. Right: Standard deviation of estimated value functions over randomized domains, averaged over 10 training seeds.

To understand what causes this difference in behavior between the two agents, we study the representations learned by the agents by analyzing the activations of the final hidden layer. We consider the agents trained on $\Xi_{big}$, and a sample of states obtained by performing a greedy rollout on a white background (which is included in $\Xi_{big}$). For each of these states, we calculate the representation corresponding to that state for another background color in $\Xi_{big}$. We then visualize these representations using t-SNE plots, where each color corresponds to a domain. A representative example of such a plot is shown in figure 5. We see that the regularized agent learns a similar representation for both backgrounds, whereas the randomized agent clearly separates them. This result indicates that the regularized agent learns to ignore the background color, whereas the randomized agent is likely to learn a different policy for a different background color. Further experiments comparing the representations of both agents can be found in the appendix.

To further study the effect of our regularization method on the representations learned by the agents, we compare the variations in the estimated value function for both agents over $\Xi_{big}$. Figure 5 shows the standard deviation of the estimated value function over different background colors, averaged over 10 training seeds and a sample of states obtained by the same procedure as described above. We observe that our regularization technique successfully reduces the variance of the value function over the randomization domain.

## 5.3 CAR RACING WITH PPO

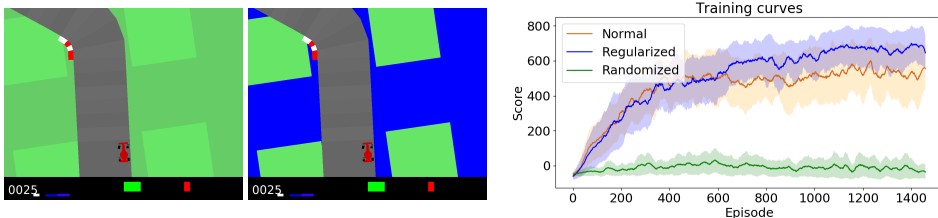

Figure 6: Left: frames from the reference and a randomized CarRacing environment. Right: training curves of our agents, averaged over 5 seeds. Shaded areas indicate the 95% confidence interval of the mean.

To demonstrate the applicability of our regularization method to other domains and algorithms, we also perform experiments with the PPO algorithm (Schulman et al., 2017) on the CarRacing environment (Brockman et al., 2016), in which an agent must drive a car around a racetrack. An example state from this environment and a randomized version in which part of the background changes color are shown in figure 6. We start by training 3 agents on this domain: a normal agent on the original background, a randomized agent, and a regularized agent with $\lambda = 50$. Randomization in this experiment occurs over the entire RGB cube, which is larger than for the cartpole experiments. Training curves are shown in figure 6. We see that the randomized agent fails to learn a successful policy on this large randomization space, whereas the other agents successfully learn.

| Agent | Return (original) | Return (all colors) |
|---|---|---|
| Normal | $554 \pm 68$ | $60 \pm 53$ |
| Randomized | $-17 \pm 33$ | $-35 \pm 9$ |
| Regularized (ours) $\lambda = 10$ | $622 \pm 81$ | $324 \pm 51$ |
| Regularized (ours) $\lambda = 50$ | $640 \pm 40$ | $553 \pm 80$ |
| Dropout 0.1 | $696 \pm 107$ | $154 \pm 86$ |
| Weight decay $10^{-4}$ | $692 \pm 55$ | $61 \pm 15$ |
| EPOpt-PPO | $-14 \pm 28$ | $-7 \pm 47$ |

Table 1: Average returns on the original environment and its randomizations over all colors, with 95% confidence intervals calculated from 5 training seeds.

We also compare the generalization ability of these agents to other agents trained with different randomization and regularization methods. On the reference domain, we train a regularized agent with a smaller value of $\lambda = 10$, and two agents respectively with dropout 0.1 and l2 weight decay of $10^{-4}$, as in Cobbe et al. (2018) and Aractingi et al. (2019). On the randomized domain, we train an agent with the EPOpt-PPO algorithm Rajeswaran et al. (2016), where in our implementation the agent only trains on the randomized domains on which its score is worse than average. Scores on both the reference domain and its randomizations are shown in 1. These results confirm that our regularization leads to agents that are both successful in training and successfully generalize to a wide range of backgrounds. Moreover, a larger value of $\lambda$ yields higher generalization scores. Of the other regularization schemes that we tested, we find that although they do improve learning on the reference domain, only dropout leads to improvement in generalization over the randomization space compared to our baseline.

## 6 CONCLUSION

In this paper we studied generalization to visually diverse environments in deep reinforcement learning. We formalized the problem, illustrated the inefficiencies of standard domain randomization, and proposed a theoretically grounded method that leads to robust, low-variance policies that generalize well. We conducted several experiments in different environments of differing complexities using both on-policy and off-policy algorithms to support our claims.

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

## A    PROOF OF PROPOSITION 1

The proof presented in the following applies to MDPs with a discrete action space. However, it can straightforwardly be generalized to continuous action spaces by replacing sums over actions with integrals over actions.

The proof uses the following lemma :

**Lemma 1** *For two distributions $p(x, y) = p(x)p(y|x)$ and $q(x, y) = q(x)q(y|x)$, we can bound the total variation distance of the joint distribution :*

$$
\begin{aligned}
D_{TV}(p(\cdot, \cdot) \| q(\cdot, \cdot)) &\leq D_{TV}(p(\cdot) \| q(\cdot)) \\
&+ \max_x D_{TV}(p(\cdot|x) \| q(\cdot|x))
\end{aligned}
$$

*Proof of the Lemma.*

We have that :

$$
\begin{aligned}
D_{TV}(p(\cdot, \cdot) \| q(\cdot, \cdot)) &= \frac{1}{2} \sum_{x,y} |p(x, y) - q(x, y)| \\
&= \frac{1}{2} \sum_{x,y} |p(x)p(y|x) - q(x)q(y|x)| \\
&= \frac{1}{2} \sum_{x,y} |p(x)p(y|x) - p(x)q(y|x) \\
&\quad + (p(x) - q(x))q(y|x)| \\
&\leq \frac{1}{2} \sum_{x,y} p(x)|p(y|x) - q(y|x)| \\
&\quad + |p(x) - q(x)|q(y|x) \\
&\leq \max_x D_{TV}(p(\cdot|x) \| q(\cdot|x)) \\
&\quad + D_{TV}(p(\cdot) \| q(\cdot))
\end{aligned}
$$

*Proof of the proposition.*

Let $p_{\phi_i}^t(s, a)$ be the probability of being in state $\phi_i(s)$ at time $t$, and executing action $a$, for $i = 1, 2$. Since both MDPs have the same reward function, we have by definition that $\eta_i = \sum_{s,a} \sum_t \gamma^t p_{\phi_i}^t(s, a) r_t(s, a)$, so we can write :

$$
\begin{aligned}
|\eta_1 - \eta_2| &\leq \sum_{s,a} \sum_t \gamma^t |p_{\phi_1}^t(s, a) - p_{\phi_2}^t(s, a)| r_t(s, a) \\
&\leq r_{\max} \sum_{s,a} \sum_t \gamma^t |p_{\phi_1}^t(s, a) - p_{\phi_2}^t(s, a)| \\
&= 2r_{\max} \sum_t \gamma^t D_{TV}(p_{\phi_1}^t(\cdot, \cdot) \| p_{\phi_2}^t(\cdot, \cdot))
\end{aligned}
\tag{3}
$$

But $p_{\phi_1}^t(s, a) = p_{\phi_1}^t(s) \pi^{\phi_1}(a|s)$ and $p_{\phi_2}^t(s, a) = p_{\phi_2}^t(s) \pi^{\phi_2}(a|s)$, Thus (Lemma 1) :

$$
\begin{aligned}
D_{TV}(p_{\phi_1}^t(\cdot, \cdot) \| p_{\phi_2}^t(\cdot, \cdot)) &\leq D_{TV}(p_{\phi_1}^t(\cdot) \| p_{\phi_2}^t(\cdot)) \\
&+ \max_s D_{TV}(\pi^{\phi_1}(\cdot|s) \| \pi^{\phi_2}(\cdot|s)) \\
&\leq D_{TV}(p_{\phi_1}^t(\cdot) \| p_{\phi_2}^t(\cdot)) \\
&+ K_\pi \|\phi_1 - \phi_2\|_\infty
\end{aligned}
\tag{4}
$$

We still have to bound $D_{TV}(p_{\phi_1}^t(\cdot)\|p_{\phi_2}^t(\cdot))$. For $s \in S$ we have that :

$$
\begin{aligned}
|p_{\phi_1}^t(s) - p_{\phi_2}^t(s)| &\leq \sum_{s'} |p_{\phi_1}(s_t = s|s')p_{\phi_1}^{t-1}(s') \\
&\quad - p_{\phi_2}(s_t = s|s')p_{\phi_2}^{t-1}(s')| \\
&= \sum_{s'} |p_{\phi_1}(s_t = s|s')p_{\phi_1}^{t-1}(s') \\
&\quad - p_{\phi_2}(s_t = s|s')p_{\phi_1}^{t-1}(s') \\
&\quad + p_{\phi_2}(s_t = s|s')p_{\phi_1}^{t-1}(s') \\
&\quad - p_{\phi_2}(s_t = s|s')p_{\phi_2}^{t-1}(s')| \\
&\leq \sum_{s'} p_{\phi_1}^{t-1}(s')|p_{\phi_1}(s|s') - p_{\phi_2}(s|s')| \\
&\quad + p_{\phi_2}(s|s')|p_{\phi_1}^{t-1}(s') - p_{\phi_2}^{t-1}(s')|
\end{aligned}
$$

Summing over $s$ we have that

$$
\begin{aligned}
D_{TV}(p_{\phi_1}^t(\cdot)\|p_{\phi_2}^t(\cdot)) &\leq \frac{1}{2} \sum_s \mathbb{E}_{s' \sim p_{\phi_1}^{t-1}}[|p_{\phi_1}(s|s') \\
&\quad - p_{\phi_2}(s|s')|] \\
&\quad + D_{TV}(p_{\phi_1}^{t-1}(\cdot)\|p_{\phi_2}^{t-1}(\cdot))
\end{aligned}
$$

But by marginalizing over actions : $p_{\phi_1}(s|s') = \sum_a \pi^{\phi_1}(a|s')p_{\phi_1}(s|a,s')$, and using the fact that $p_{\phi_1}(s|a,s') = T_{\phi_1}(s|a,s') = T_{\phi_2}(s|a,s') = p_{\phi_2}(s|a,s') := p(s|a,s')$, we have that

$$
\begin{aligned}
|p_{\phi_1}(s|s') - p_{\phi_2}(s|s')| &= |\sum_a p(s|a,s')(\pi^{\phi_1}(a|s') \\
&\quad - \pi^{\phi_2}(a|s'))| \\
&\leq \sum_a p(s|a,s')|\pi^{\phi_1}(a|s') \\
&\quad - \pi^{\phi_2}(a|s')|
\end{aligned}
$$

And using $\sum_s p(s|a,s') = 1$ we have that :

$$
\begin{aligned}
&\frac{1}{2} \sum_s \mathbb{E}_{s' \sim p_{\phi_1}^{t-1}}[|p_{\phi_1}(s|s') - p_{\phi_2}(s|s')|] \\
&\leq \frac{1}{2}\mathbb{E}_{s' \sim p_{\phi_1}^{t-1}} \sum_a [\sum_s p(s|a,s')]|\pi^{\phi_1}(a|s') - \pi^{\phi_2}(a|s')| \\
&\leq \max_{s'} D_{TV}(\pi^{\phi_1}(\cdot|s)\|\pi^{\phi_2}(\cdot|s)) \\
&\leq K_\pi \|\phi_1 - \phi_2\|_\infty
\end{aligned}
$$

Thus, by induction, and assuming $D_{TV}(p_{\phi_1}^0(\cdot)\|p_{\phi_2}^0(\cdot)) = 0$ :

$$
D_{TV}(p_{\phi_1}^t(\cdot)\|p_{\phi_2}^t(\cdot)) \leq t K_\pi \|\phi_1 - \phi_2\|_\infty
$$

Plugging this into inequality 4, we get

$$
D_{TV}(p_{\phi_1}^t(\cdot,\cdot)\|p_{\phi_2}^t(\cdot,\cdot)) \leq (t+1) K_\pi \|\phi_1 - \phi_2\|_\infty
$$

We also note that the total variation distance takes values between 0 and 1, so we have

$$
D_{TV}(p_{\phi_1}^t(\cdot,\cdot)\|p_{\phi_2}^t(\cdot,\cdot)) \leq \min(1, (t+1) K_\pi \|\phi_1 - \phi_2\|_\infty)
$$

Plugging this into inequality 3 leads to our first bound,

$$
|\eta_1 - \eta_2| \leq 2r_{\max} \sum_t \gamma^t \min(1, (t+1) K_\pi \|\phi_1 - \phi_2\|_\infty)
$$

Our second, looser bound can now be achieved as follows,

$$|\eta_1 - \eta_2| \leq 2r_{\max} \sum_t \gamma^t(t+1) K_\pi \|\phi_1 - \phi_2\|_\infty$$

$$|\eta_1 - \eta_2| \leq \frac{2r_{\max}}{(1-\gamma)^2} K_\pi \|\phi_1 - \phi_2\|_\infty$$

## B  COMPARISON OF OUR WORK WITH ARACTINGI ET AL. (2019)

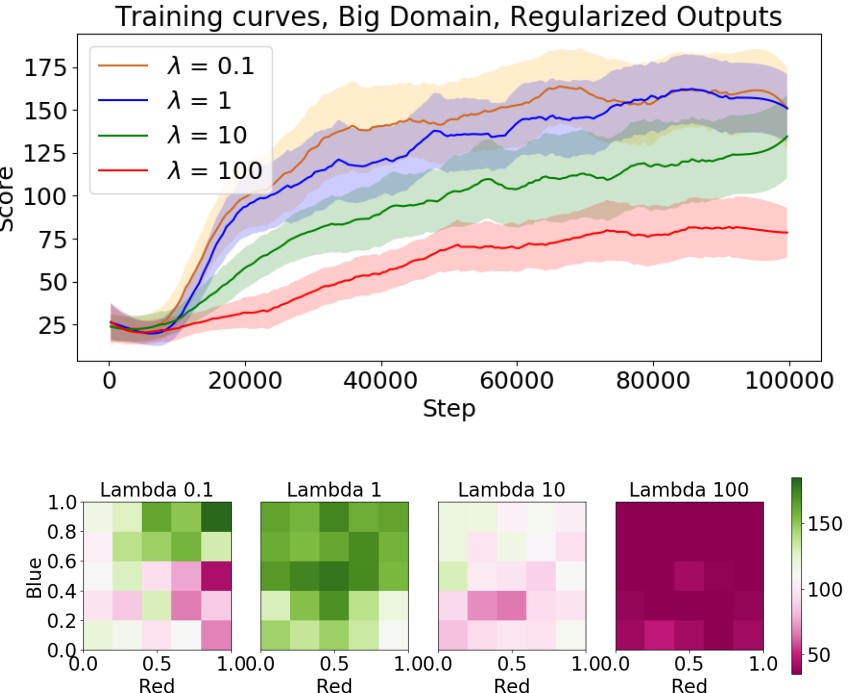

Figure 7: Top: training curves of agents with different regularization strengths when following the scheme of Aractingi et al. (2019). Shaded errors correspond to 95% confidence intervals of the mean, calculated from 10 training seeds. Bottom: scores obtained by trained agents for different regularization strengths on a plane within the RGB cube.

Concurrently and independently of our work, Aractingi et al. (2019) propose a similar regularization scheme on randomized visual domains, which they experimentally demonstrate with the PPO algorithm on the VizDoom environment Kempka et al. (2016) with randomized textures. As opposed to the regularization scheme proposed in our work in which we regularize the final hidden layer of the network, they propose regularizing the output of the policy network. Regularizing the last hidden layer as in our scheme more clearly separates representation learning and policy learning, since the final layer of the network is only affected by the RL loss.

We hypothesized that regularizing the output of the network directly could lead to the regularization loss and the RL loss competing against each other, such that a tradeoff between policy performance and generalization would be necessary. To test this hypothesis, we performed experiments on the visual cartpole domain with output regularization with different values of regularization parameter $\lambda$. Our results are shown in figure 7. We find that increasing the regularization strength adversely affects training. However, agents trained with higher values of $\lambda$ do achieve more consistent results over the randomization space. This shows that there is indeed a tradeoff between generalization and policy performance when regularizing the network output as in Aractingi et al. (2019). In our experiments, however, we have found that changing the value of $\lambda$ only affects generalization ability and not agent performance on the reference domain.

## C  EXPERIMENTAL DETAILS

All code used for our experiments will be made available online.

### C.1  STATE PREPROCESSING

For our implementation of the visual cartpole environment, each image consists of $84 \times 84$ pixels with RGB channels. To include momentum information in our state description, we stack $k = 3$ frames, so the shape of the state that is sent to the agent is $84 \times 84 \times 9$.

We note that because of this preprocessing, agents trained until convergence achieve average returns of about 175 instead of the maximum achievable score of 200. Since the raw pixels do not contain momentum information, we stack three frames as input to the network. When the environment is reset, two random actions are thus taken before the agent is allowed to make a decision. For some initializations, this causes the agent to start in a situation it cannot recover from. Moreover, due to the low image resolution the agent may sometimes struggle to correctly identify momentum and thus may make mistakes.

In CarRacing, each state consists of $96 \times 96$ pixels with RGB channels. We introduce frame skipping as is often done for Atari games (Mnih et al. (2015)), with a skip parameter of 5. This restricts the length of an episode to 200 action choices. We then stack 2 frames to include momentum information into the state description. The shape of the state that is sent to the agent is thus $96 \times 96 \times 6$. We note that although this preprocessing makes training agents faster, it also causes trained agents to not attain the maximum achievable score on this environment.

### C.2  VISUAL CARTPOLE

#### C.2.1  EXTRAPOLATION

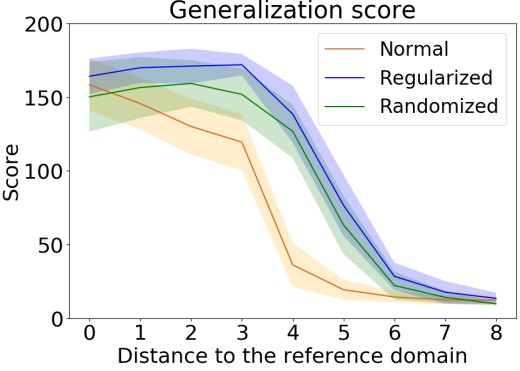

Figure 8: Generalization scores, with 95% confidence intervals obtained over 10 training seeds. The normal agent is trained on white $(1, 1, 1)$, corresponding to a *distance to train*$= 0$. The rest of the domains correspond to $(x, x, x)$, for $x = 0.9, 0.8, \ldots, 0$.

Given that regularized agents are stronger in interpolation over their training domain, it is natural to wonder what the performance of these agents is in extrapolation to colors not within the range of colors sampled within training. For this purpose, we consider randomized and regularized agents trained on $\Xi_{big}$, and test them on the set $\{(x, x, x), 0 \leq x \leq 1\}$. None of these agents was ever exposed to $x \leq 0.5$ during training.

Our results are shown in figure 8. We find that although the regularized agent consistently outperforms the randomized agent in interpolation, both agents fail to extrapolate well outside the train domain. Since we only regularize with respect to the training space, there is indeed no guarantee that our regularization method can produce an agent that extrapolates well. Since the objective of domain randomization often is to achieve good transfer to an a priori unknown target domain, this

result suggests that it is important that the target domain lie within the randomization space, and that the randomization space be made as large as possible during training.

### C.2.2 FURTHER STUDY OF THE REPRESENTATIONS LEARNED BY DIFFERENT AGENTS

We perform further experiments to demonstrate that the randomized agent learns different representations for different domains, whereas the regularized agent learns similar representations. We consider agents trained on $\Xi_{split} = [0, 0.2]^3 \cup [0.8, 1]^3$, the union of *darker*, and *lighter* backgrounds. We then rollout each agent on a single episode of the domain with a white background and, for each state in this episode, calculate the representations learned by the agent for other background colors. We visualize these representations using the t-SNE plot shown in figure 9. We observe that the randomized agent clearly separates the two training domains, whereas the regularized agent learns similar representations for both domains.

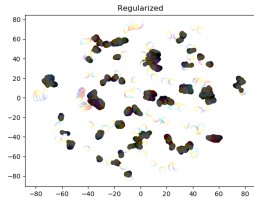 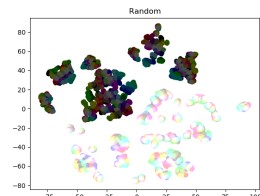

Figure 9: t-SNE of the representations over $\Xi_{split}$ of the Regularized (Left) and Randomized (Right) agents. Each color corresponds to a domain. The randomized agent learns very different representations for $[0, 0.2]^3$ and $[0.8, 1]^3$.

We are interested in how robust our agents are to unseen values $\xi \notin \Xi_{split}$. To visualize this, we rollout both agents in domains having different background colors : $\{(x, x, x), 0 \le x \le 1\}$, i.e ranging from black to white, and collect their features over an episode. We then plot the t-SNEs of these features for both agents in figure 10, where each color corresponds to a domain.

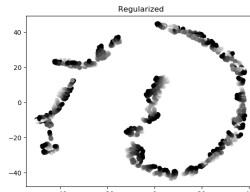 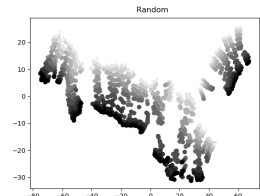

Figure 10: t-SNE of the features of the Regularized (Left) and Randomized (Right) agents. Each color corresponds to a domain.

We observe once again that the regularized agent has much lower variance over unseen domains, whereas the randomized agent learns different features for different domains. This shows that the regularized agent is more robust to domain shifts than the randomized agent.

## D FURTHER RELATED WORK: LIPSCHITZ CONTINUITY AND GENERALIZATION IN DEEP LEARNING

Lipschitz-sensitive bounds on the generalization abilities of neural networks have a long history (Bartlett (1997); Anthony & Bartlett (2009); Neyshabur et al. (2015)). Recently, Bartlett et al. (2017) proved a generalization bound in terms of the norms of each layer, which is proportional to the Lipschitz constant of the network. Oberman & Calder (2018) studied generalization through a general empirical risk minimization procedure with Lipschitz regularization, and provides generalization bounds. Similarly, we show that the Lipschitz constant of the network with respect to the randomization parameters plays an important role in achieving zero-shot transfer to a target domain.

# E    ALGORITHMS

---

**Algorithm 1** Deep Q-learning with our regularization method

---

Initialize replay memory $\mathcal{D}$ to capacity $N$
Initialize action-value function $Q$ with random weights $\theta$
Initialize the randomization space $\Xi$, and a reference MDP $M_{\phi^{ref}}$ to train on.
Initialize a regularization parameter $\lambda$
Define a feature extractor $f_\theta$
**for** episode = 1, $M$ **do**
    Sample a randomizer function $\phi^{sampled}$ uniformly from $\Xi$.
    **for** $t = 1, T$ **do**
        With probability $\epsilon$ select a random action $a_t$
        otherwise select $a_t = \arg\max_a Q(\phi^{ref}(s_t), a; \theta)$
        Execute action $a_t$ in $M_{\phi^{ref}}$, observe reward $r_t$, the reference state, and the corresponding
        randomized state with the chosen visual settings: $\phi^{ref}(s_{t+1}), \phi^{sampled}(s_{t+1})$
        Store transition $(\phi^{ref}(s_t), \phi^{sampled}(s_t), a_t, r_t, \phi^{ref}(s_{t+1}), \phi^{sampled}(s_{t+1}))$ in $\mathcal{D}$
        Sample random minibatch of transitions $(\phi_j^{ref}, \phi_j^{sampled(old)}, a_j, r_j, \phi_{j+1}^{ref}, \phi_{j+1}^{sampled(old)})$
        from $\mathcal{D}$, where $\phi_j^{sampled(old)}$ is the randomization that had been selected when the transi-
        tion had been observed.
        Set $y_j = \begin{cases} r_j \text{ for terminal } \phi_{j+1}^{ref}. \\ r_j + \gamma \max_{a'} Q(\phi_{j+1}^{ref}, a', \theta) \\ \text{otherwise.} \end{cases}$
        Perform a gradient descent step on $(y_j - Q(\phi_j^{ref}, a_j; \theta))^2 + \lambda \|f_\theta(\phi_j^{ref}) - f_\theta(\phi_j^{sampled(old)})\|_2^2$
    **end for**
**end for**

---

---

**Algorithm 2** Policy Gradient with a baseline using our regularization method

---

Initialize policy network function $\pi_\theta$ with random weights $\theta$, baseline $b_\theta$
Initialize the randomization space $\Xi$, and a reference MDP $M_{\phi^{ref}}$ to train on.
Initialize a regularization parameter $\lambda$
Define a feature extractor $f_\theta$
**for** episode $= 1, M$ **do**
    Collect a set of trajectories $(s_0, a_0, r_1, \ldots, s_{T-1}, a_{T-1}, r_T)$ by executing $\pi_\theta$ on $M_{\phi^{ref}}$.
    For each trajectory, sample a randomizer function $\phi^{sampled}$ uniformly from $\Xi$.
    **for** $t = 1, T$ in each trajectory **do**
        Compute the return $R_t = \sum_{t'=t}^{T-1} \gamma^{t'-t} r_{t'}$
        Estimate the advantage $\hat{A}_t = R_t - b_\theta(\phi^{ref}(s_t))$
    **end for**
    Perform a gradient descent step on

$$\sum_{t=0}^{T} \big[ - \hat{A}_t \log \pi_\theta(a_t | \phi^{ref}(s_t))$$
$$+ \| R_t - b_\theta(\phi^{ref}(s_t)) \|_2^2$$
$$+ \lambda \| f_\theta(\phi^{ref}(s_t)) - f_\theta(\phi^{sampled}(s_t)) \|_2^2 \big]$$

**end for**

---

Note that algorithm 2 can be straightforwardly adapted to several state of the art policy gradient algorithms such as PPO.

## F   A DYNAMICS RANDOMIZATION EXPERIMENT

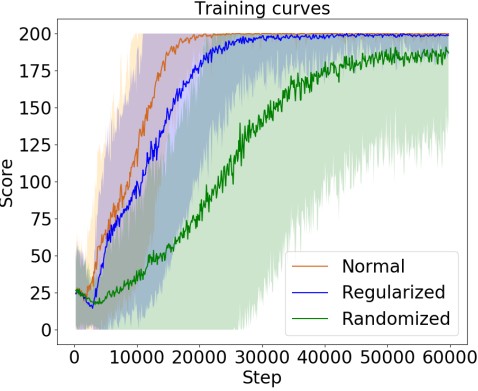

Figure 11: Training curves of the agents on the Cartpole domain, averaged over 100 seeds.

We also perform an experiment to demonstrate that learning domain-invariant features can be helpful not only for visual randomization, but also for some instances of dynamics randomization. We consider once again the Cartpole domain, where this time we randomize some physical dynamics of the environment. Specifically, we choose to randomize the pole length $l$, and the gravity $g$. The state for our reinforcement learning agent is the 6-dimensional vector $(x, \dot{x}, \theta, \dot{\theta}, l, g)$, where $x$ is the position of the cart, $\dot{x}$ its velocity, $\theta$ the angle between the pole and the horizontal plane, and $\dot{\theta}$ the angular velocity. We train all three agents (Normal, Regularized, Randomized) using only two randomized domains : $l = 0.5, g = 9.8$ and $l = 1, g = 50$. We use the DQN algorithm with a Multi-Layer Perceptron architecture for this experiment.

We first compare the training speed of the agents. The training curves averaged over 100 seeds are plotted in figure 11. We observe once again that the randomized agent is significantly slower than the regularized one, and is more unstable.

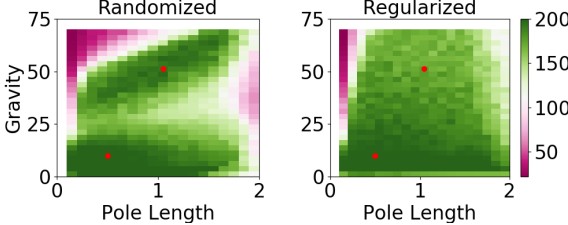

Figure 12: Generalization scores averaged over 5 training seeds and 4 test episodes per seed. Red dots correspond to training environments

Next, we examine the agents' generalization ability. We test the agents on environments having values of pole length $l$ and gravity $g$ unseen during training. We plot their scores in figure 12. The randomized agent clearly specializes on the two different training domains, corresponding to the two clearly distinguishable regions where high scores are achieved, whereas the regularized agents achieves more consistent scores across domain. This result can be understood as follows. Although the different dynamics between the two domains lead to there being different sets of optimal policies, our regularization method forces the agent to only learn policies that do not depend on the specific values of the randomized dynamics parameters. These policies are therefore more likely to also work when those dynamics are different.

### F.1 REPRESENTATIONS LEARNED BY THE AGENT

We analyze the representations learned by each agent in our dynamics randomization experiment. Once the agents are trained, we rollout their policies in both randomized environments with an $\epsilon$-greedy strategy, where we use $\epsilon = 0.2$ to reach a larger number of states of the MDP, over 10000 steps. We collect the representations (the activations of the last hidden layer) corresponding to the visited states. These features are 100-dimensional, so in order to visualize them, we use the t-SNE plots shown in figure 13. We emphasize that although this figure corresponds to a single training seed, the general aspect of these results is repeatable.

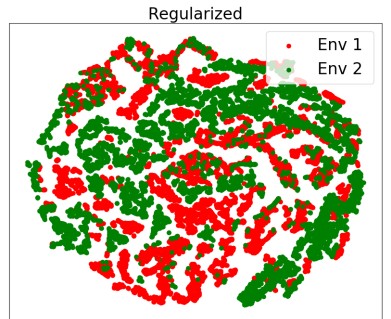
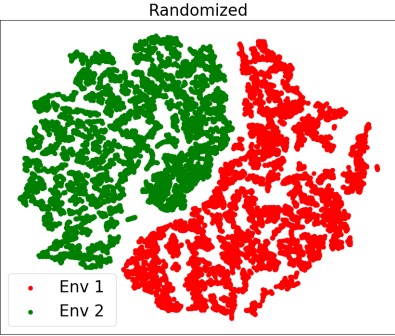

Figure 13: t-sne of the representations learned by the regularized and randomized agents on the two training environments.

The randomized agent learns completely different representations for the two randomized environments. This explains its high variance during the training, since it tries to learn a different strategy for each domain. On the other hand, our regularized agent has the same representation for both domains, which allows it to learn much faster, and to learn policies that are robust to changes in the environment's dynamics.

