# OpenReview forum: "Robust Domain Randomization for Reinforcement Learning"
_ICLR.cc/2020/Conference — Reject_

### Official Review · AnonReviewer3 · 2019-10-10
**Official Blind Review #3**

**Rating:** 3

**Review:**

The paper introduces the high variance policies challenge in domain randomization for reinforcement learning. The paper gives a new bound for the expected return of the policy when the policy is Lipschitz continuous. Then the paper proposes a new method to minimize the Lipschitz constant for policies of all randomization. Experiment results prove the efficacy of the proposed domain randomization method for various reinforcement learning approaches.

The paper mainly focuses on the problem of visual randomization, where the different randomized domains differ only in state space and the underlying rewards and dynamics are the same. The paper also assumes that there is a mapping from the states in one domain to another domain. Are there any constraints on the mapping? Will some randomization introduces a larger state space than others?

The paper demonstrates that the expected return of the policy is bounded by the largest difference in state space and the Lipschitz constant of the policies, which is a new perspective of domain randomization for reinforcement learning.

The proposed method minimizes the expected variations between states of two randomizations but the Lipschitz constant is by the largest difference of policy outputs of a state pair between domains. Should minimizing the maximum difference be more proper?

The center part of Figure 2 is confusing, could the authors clarify it?

In the Grid World environment, how does the random parameter influence the states?

The baselines are a little weak. The paper only compares the proposed with training reinforcement learning algorithm on randomized environments. Could the authors compare with other domain randomization methods in reinforcement learning or naively adapt domain randomization methods from other areas to reinforcement learning?

Overall, the paper is well-written and the ideas are novel. However, some parts are not clearly clarified and the experiments are a little weak with too weak baselines. I will consider raising my score according to the rebuttal.

Post-feedback:
I have read the rebuttal. The authors have addressed some of my concerns but why minimizing the expected difference is not convincing. I think the paper should receive a borderline score between 3 and 6.

**Experience Assessment:**

I have read many papers in this area.

**Review Assessment: Checking Correctness Of Derivations And Theory:**

I carefully checked the derivations and theory.

**Review Assessment: Checking Correctness Of Experiments:**

I carefully checked the experiments.

**Review Assessment: Thoroughness In Paper Reading:**

I read the paper at least twice and used my best judgement in assessing the paper.

---

> ### Author Response · Authors · 2019-11-09
> **Response to Reviewer 3**
>
> We thank reviewer 3 for their effort and for their feedback. We respond to their comments as follows, and have correspondingly updated our manuscript:
>
> 1) For the Lipschitz constant of the network to be finite, two different randomizations must not lead to the same state (otherwise the denominator in definition 2 would be zero). Each randomization conserves the same number of states in the MDP, however the randomization space may be bigger for some randomizations than others.
>
> 2) This is an interesting idea, which we had not considered. Minimizing the maximum difference would seem quite impractical though, since after each gradient descent step we would have to re-identify the state and the randomization that maximizes the difference.
>
> 3) We used the following procedure to make figure 2 (center). For a given value of lambda, we train a regularized agent on the reference domain. We then measure the difference in returns obtained by this agent on the reference and on the randomized domain, and this return determines the agent’s position along the x axis. We then numerically calculate the Lipschitz constant from the agent’s action distribution over all states, and use this constant to calculate the bound in proposition 1. This bound determines the agent’s position along the y axis. This process is repeated for 20 random seeds per value of lambda.
>
> We have updated the description of figure 2 both in the caption and in the main text to make this clearer.
>
> 4) The state observed by the agent is a vector of size 3, where the first two coordinates are x and y and the third coordinate directly is the randomization parameter. Since the agent is parameterized by a neural network that takes the state as input, different values of the randomization parameter lead to different probabilities over actions. We have now rephrased the relevant section in the main text to now refer to the (x,y,xi) tuple as the state that is observed by the agent.
>
> 5) Following suggestions from reviewers 1 and 2, we have performed experiments comparing our method both to regularization methods studied in [Cobbe 19] (dropout and weight decay) and to the EPOpt-PPO algorithm [Rajeswaran' 16]. We have included our results in table 1, and have found that although dropout and weight decay do help in training over the reference domain, only dropout improves the agents’ generalization ability. However, the generalization score of the dropout agent remains significantly lower than that of agents trained with our regularization method.  We also found that EPOpt-PPO was ineffective in this setting.
>
> [Cobbe' 19] Cobbe, K., Klimov, O., Hesse, C., Kim, T. and Schulman, J., Quantifying generalization in reinforcement learning. In ICML, 2019.
>
> [Rajeswaran' 16]  Rajeswaran, Aravind, et al. "Epopt: Learning robust neural network policies using model ensembles." arXiv preprint arXiv:1610.01283 (2016)

---

### Official Review · AnonReviewer2 · 2019-10-19
**Official Blind Review #2**

**Rating:** 3

**Review:**

Summary:

To improve the generalization ability of deep RL agents across the tasks with different visual patterns, this paper proposed a simple regularization technique for domain randomization. By regularizing the outputs from normal and randomized states, the trained agents are forced to learn invariant representations. The authors showed that the proposed method can be useful to improve the generalization ability using CartPole and Car Racing environments.

Detailed comments:

I'd like to recommend "weak reject" due to the following reasons:

1. Lack of novelty: The main idea of this paper (i.e. regularizing the outputs from normal and randomized states) is not really new because it has been explored before [Aractingi' 19]. Even though this paper provides more justification and analysis for this part (Proposition 1 in the draft), the contributions are not enough as the ICLR publications.

2. As shown in [Cobbe' 19], various regularization and data augmentation techniques have been studied for improving the generalization ability of deep RL agents. Therefore, the comparisons with such baselines are required to verify the effectiveness of the proposed methods.

3. For domain randomization, it has been observed that finding a good distribution of simulation parameters is a key component [Ramos' 19, Mozifian' 19, Chebotar' 19], but the authors did not consider training the distribution of simulation parameters in the paper.

[Ramos' 19] Ramos, F., Possas, R.C. and Fox, D., BayesSim: adaptive domain randomization via probabilistic inference for robotics simulators. In RSS, 2019.

[Cobbe' 19] Cobbe, K., Klimov, O., Hesse, C., Kim, T. and Schulman, J., Quantifying generalization in reinforcement learning. In ICML, 2019.

[Mozifian' 19] Mozifian, M., Higuera, J.C.G., Meger, D. and Dudek, G., Learning Domain Randomization Distributions for Transfer of Locomotion Policies. arXiv preprint arXiv:1906.00410, 2019.

[Chebotar' 19] Chebotar, Y., Handa, A., Makoviychuk, V., Macklin, M., Issac, J., Ratliff, N. and Fox, D., May. Closing the sim-to-real loop: Adapting simulation randomization with real world experience. In 2019 International Conference on Robotics and Automation (ICRA) (pp. 8973-8979). 2019

[Aractingi' 19] Michel Aractingi, Christopher Dance,  Julien Perez, Tomi Silander,  Improving the Generalization of Visual Navigation Policies using Invariance Regularization, ICML workshop 2019.

**Experience Assessment:**

I have read many papers in this area.

**Review Assessment: Checking Correctness Of Derivations And Theory:**

I carefully checked the derivations and theory.

**Review Assessment: Checking Correctness Of Experiments:**

I carefully checked the experiments.

**Review Assessment: Thoroughness In Paper Reading:**

I read the paper thoroughly.

---

> ### Author Response · Authors · 2019-11-09
> **Response to Reviewer 2**
>
> We thank reviewer 2 for their effort and for their feedback. We respond to their comments as follows, and have correspondingly updated our manuscript:
>
> 1) We thank reviewer 2 for pointing out this reference, which we were unaware of. [Aractingi' 19] propose a regularization method that is indeed similar to ours. However, we feel that this does not detract from the novelty of our work. From the date at which [Aractingi' 19] appeared on OpenReview, it appears that their work is concurrent to ours. Moreover, their work does not yet seem to be available through the standard dissemination channels (conference proceedings, journal publication, or arxiv). As such, we believe that both [Aractingi' 19] and our work can stake an equal claim to novelty.
>
> We would also like to point out some important differences between their work and ours. As mentioned by reviewer 2, we do provide more justification and analysis. We also demonstrate our scheme with a value-based algorithm, whereas they test theirs only with PPO. More importantly, our regularization scheme avoids a significant shortcoming of their proposal. While we regularize the final hidden layer of the network, they regularize the network output. Regularizing the network output as opposed to an intermediate layer has the potential of causing a tradeoff between policy learning and generalization.
>
> To test whether such a tradeoff exists in their scheme, we have now performed a new experiment (discussed in the appendix) in which we regularize the output layer as in [Aractingi' 19] with several regularization strengths of the DQN agent in cartpole. We show that strong regularizations lead to poor scores but more stable performance across domains, while weak regularizations lead to higher scores but worse generalizations. Our regularization scheme allows us to achieve both objectives.
>
> In addition to this new section in the appendix, we have also added a paragraph to the related works section in the main text discussing [Aractingi' 19].
>
> 2) Following this suggestion, we have now performed these experiments and have added our results to table 1, as well as a discussion in the main text. We find that although dropout and weight decay do help in training over the reference domain, only dropout improves the agents’ generalization ability. However, the generalization score of the dropout agent remains significantly lower than that of agents trained with our regularization method. We have also added a comparison to EPOpt-PPO, which we found to be ineffective.
>
> 3) Both [Ramos 19] and [Chebotar 19] require real world data to adapt the distribution of simulation parameters, whereas we attempt to achieve good results over as large a distribution as possible without real world data. [Mozifian 19] deals with dynamics randomization, in which some simulation parameter values lead to different scores being achievable. We feel that these are outside the scope of the paper; however, as suggested by reviewer 1 the scope of the paper was not properly detailed in our work. We have thus changed the abstract, introduction, and conclusion to better reflect the scope.

---

### Official Review · AnonReviewer1 · 2019-10-24
**Official Blind Review #1**

**Rating:** 3

**Review:**

This paper proposes a regularization scheme for training vision-based control policies that are robust to variations of the visual input, which the paper classifies as "visual randomization". Visual randomization, as the paper notes, is one of the state-of-the-art techniques proposed from simulation to real robot transfer of vision-based policies trained on synthetic images. The regularization proposed in the paper aims to produce policies that rely on features that are invariant to the visual randomization, so that the resulting behaviour of the agent is consistent across different variations of its visual input for the same states.

The paper proposes that forcing a policy trained under randomization to have a Lipschitz constant of K, over the randomization parameters, causes the optimal policy to be similar across randomization parameters, with the difference in expected returns of two randomized environments being bounded by K times the maximum difference in their randomization parameters.

I recommend this paper to not be accepted until the following issues are addressed.

* There are missing details from the experimental setup, which makes the results hard to interpret (see, below).

* There are  missing details on how vanilla domain randomization was implemented. Domain randomization aims to maximize the expected performance over the environment distribution. This can be implemented properly by computing the expected gradients with data from more than one environment. From the algorithm descriptions in the appendix, it is not clear that this is how vanilla domain randomization was implemented.

* The title, introduction and conclusions do not reflect the scope of the paper. The paper only addresses situations where the same behaviour is achievable on all environments, an assumption (Mehta et al, 2019) also makes, and its proposed regularization is based on the assumption that the optimal behaviour is achievable with the same policy on all environments. But this is not true in general: for dynamics randomization, different environments may require different policies (e.g. driving a car on a road vs driving off-road). The regularization method may result in overly conservative policies i such situations.

Questions about experimental details:

What are the maximum returns  for Cartpole when trained until convergence without randomization? (175? 200? 1000?) If the maximum returns are higher than 175, how does Figure 4 look with more data? This is crucial to understand, for example, the results in Figure 11. That figure shows the proposed regularization slightly hinders the performance for the environments near l=1, g=50 (that region is a darker shade of green on the left subfigure). How do we know if the task has been successfully solved in the green vs purple regions? In all experiments, are the training curves showing the performance of the policies over the same environments (same seeds)? If not, how are the training curves comparable?

Other things to improve:

The conclusions of this paper can be made stronger by adding a comparison with EpOpt-PPO (i.e. optimizing the worst case performance over a set of trajectories sampled from multiple environments)

**Experience Assessment:**

I have published one or two papers in this area.

**Review Assessment: Checking Correctness Of Derivations And Theory:**

I assessed the sensibility of the derivations and theory.

**Review Assessment: Checking Correctness Of Experiments:**

I carefully checked the experiments.

**Review Assessment: Thoroughness In Paper Reading:**

I read the paper thoroughly.

---

> ### Author Response · Authors · 2019-11-09
> **Response to Reviewer 1**
>
> We thank reviewer 1 for their effort and for their feedback. We respond to their comments as follows, and have correspondingly updated our manuscript:
>
> 1) Details about the experimental setup: the maximum theoretical returns for Cartpole are 200, however in our case agents trained until convergence achieve average returns of about 175, which we believe is caused by a combination of frame stacking and low image resolution. Since the raw pixels do not contain momentum information, we stack three frames as input to the network. When the environment is reset, two random actions are thus taken before the agent is allowed to make a decision. For some initializations, this causes the agent to start in a situation it cannot recover from. Moreover, due to the low image resolution the agent may sometimes struggle to correctly identify momentum and thus may make mistakes. We have added this information to the appendix.
>
> In each experiment, background colors are randomly sampled at the start of each episode with different random seeds. However, we find that within each class of agent different random seeds all yield similar training curves. Moreover, we perform experiments with multiple seeds per agent (10 for Cartpole and 5 for Car Racing), and all our figures represent 95% confidence intervals of the training curves over these different seeds, which ensures that our training curves are comparable.
>
> As for the dynamics randomization experiment presented in the appendix and figure 11, as mentioned by reviewer 1 in dynamics randomization experiments different randomizations may indeed require different policies, which may explain why our regularization leads to worse results in some parts of randomization space.
>
> 2) There indeed was missing information as to how we implemented vanilla domain randomization, and have updated the algorithm descriptions in the appendix. We confirm that we take expected gradients with data from several environments in both value-based and policy-based algorithms.
>
> 3) We have now updated the title, introduction, and conclusion to better reflect the scope of the paper. In particular, we have changed the title to “Robust Visual Domain Randomization for Reinforcement Learning”. Several other modifications have been made throughout to clearly state that we mostly deal with visual differences in the environment.
>
> 4) Following this suggestion, we have now performed a comparison to EPOpt-PPO, and have included our results in table 1. However, we find that since the agent that uses vanilla domain randomization is unable to learn a good policy at all on car racing, optimizing over the subset of environments on which it has bad performance does not yield good results either. As requested by the other reviewers, we have also performed additional experiments in which we compare our regularization method to some other regularization methods studied for reinforcement learning in [Cobbe' 19].
>
> [Cobbe' 19] Cobbe, K., Klimov, O., Hesse, C., Kim, T. and Schulman, J., Quantifying generalization in reinforcement learning. In ICML, 2019.

---

### Decision · Program_Chairs · 2019-12-19

**Decision:**

Reject

**Comment:**

The paper presents a technique for learning RL agents to generalize well to unseen environments.

All reviewers and AC think that the paper has some potential but is a bit below the bar to be accepted due to the following facts:

(a) Limited experiments, i.e., consider more appealing baselines/scenarios and provide more experimental details.
(b) The proposed method/idea is simple/reasonable, but not super novel, i.e., not enough considering the ICLR high standard (potentially enough for a workshop paper).

Hence, I think this is a borderline paper toward rejection.